# Effects of Acute Exercise on Cognitive Flexibility in Young Adults with Different Levels of Aerobic Fitness

**DOI:** 10.3390/ijerph19159106

**Published:** 2022-07-26

**Authors:** Beibei Shi, Hong Mou, Shudong Tian, Fanying Meng, Fanghui Qiu

**Affiliations:** 1Department of Physical Education, Qingdao University, Qingdao 266071, China; shibeibei1107@163.com (B.S.); mouhong2021@126.com (H.M.); tsdtyxy@163.com (S.T.); 2Institute of Physical Education, Huzhou University, Huzhou 313000, China; 02730@zjhu.edu.cn

**Keywords:** HIIE, MICE, cognitive flexibility, aerobic fitness, more-odd shifting task

## Abstract

This study aimed to evaluate the effects of high-intensity interval exercise (HIIE) and moderate-intensity continuous exercise (MICE) on cognitive flexibility in young adults with differing levels of aerobic fitness. Sixty-six young adults were grouped into high- and low-fit groups based on their final running distance on the 20 m Progressive Aerobic Cardiovascular Endurance Run (PACER) test. Individuals participated in a 10 min HIIE, a 20 min HIIE, a 20 min MICE, and a control session (reading quietly in a chair) in a counterbalanced order. The more-odd shifting task was completed before and approximately 5 min after each intervention to assess cognitive flexibility. The results showed that young adults with a high fitness level gained greater benefits in terms of switch cost from the 20 min HIIE, while low-fitness participants benefited more from the 10 min HIIE and the 20 min MICE. These findings suggest that aerobic fitness may influence the effect of acute HIIE and MICE on cognitive flexibility. Young adults should consider individual fitness level when adopting time-effective and appropriate exercise routines to improve cognitive flexibility.

## 1. Introduction

Cognitive flexibility is an important component of executive functioning that refers to the ability to quickly alter cognition when switching between different tasks [1,2]. Cognitive flexibility is predictive of social competence and plays an essential role in problem-solving, emotion regulation, and academic performance [3,4,5]. Growing evidence has demonstrated the beneficial effects of acute exercise, also known as a single bout of exercise, on cognition [6]. Most prior research has focused on the effect of acute exercise on inhibitory control and working memory. Studies using task switching to examine cognitive flexibility have reported benefits of acute exercise in young adults [7,8]. However, a few studies have failed to detect an exercise effect on cognitive flexibility after 40 min of cycling [9,10], highlighting the importance of exploring possible moderating factors. Previous researchers have identified that the modality of acute exercise and the aerobic fitness level of study participants are crucial factors in the relationship between acute exercise and cognition [11,12,13,14].

It has been proposed that exercise modality modulates exercise-induced benefits on cognition [12,15,16]. Most studies have used moderate-intensity continuous exercise (MICE) to explore the effects of acute exercise on cognitive flexibility [7,17,18], and the majority of the results have shown improvements in cognitive flexibility after MICE [8,17,19,20]. The meta-analysis by Oberste et al. (2019) suggested that there was no difference in the cognitive benefits of continuous exercise of different intensities (high, moderate and low) [21]. However, the inverted-U hypothesis suggests that moderate intensity exercise has a greater effect on cognition than low- and high-intensity exercise [12]. High-intensity interval exercise (HIIE) was defined as repetitive high-intensity exercise interspersed with recovery periods of light intensity. In recent years, HIIE has received considerable interest as a viable alternative to high-intensity continuous exercise due to its similar cognitive benefits and lower overall exercise volume compared to high-intensity continuous exercise [21,22]. To further investigate the optimal exercise modality for improving executive function, some researchers compared HIIE and MICE and found that the effects of HIIE on working memory and inhibitory control were superior to those of MICE [23,24,25]. However, to our knowledge, to date, only one study has compared the effects of HIIE and MICE on cognitive flexibility, and the results showed greater improvement in cognitive flexibility after HIIE than MICE [26]. Therefore, the influences of different modalities of acute exercise on cognitive flexibility need further investigation.

Another possible moderator in the relationship between exercise and cognitive flexibility that has received increasing attention is aerobic fitness. Higher levels of fitness have been associated with enhancements in brain structure, brain functioning, and cognitive performance [13,27]. Aerobic fitness may act as a moderator of the effects of acute exercise on cognition and may also interact with exercise intensity, as individuals with higher fitness are accustomed to higher workloads and perceive exercise-induced physical stress differently than those who do not regularly train [28]. Notably, some experimental studies have observed a greater positive effect of acute aerobic exercise on executive function in high-fitness individuals than in low-fitness individuals [29,30,31]. For example, Tsai et al. (2016) explored the effect of MICE on cognitive flexibility in high- and low-fitness young adults using a more-odd shifting task and found that only high-fitness participants gained behavioral and neurophysiological benefits from acute aerobic exercise [32]. Similarly, some studies in older adults have found greater benefits from MICE for inhibitory control in high rather than low fitness individuals [30,31]. In contrast, several studies assessing working memory [33] and inhibitory control [34] after acute exercise in young adults with high and low aerobic fitness have only found that low health individuals benefit from acute exercise. Due to discrepancies in the existing literature regarding the age of the individuals and the cognitive tasks tested, it is not possible to establish what role aerobic fitness specifically plays in the relationship between acute exercise and cognitive flexibility. Furthermore, the majority of current research has used MICE as an intervention, and there has been little exploration of cognitive flexibility in individuals with different aerobic fitness in HIIE, making it unclear whether aerobic fitness plays a similar role in HIIE and MICE.

The purpose of the current study was to investigate whether there are differences in the effect of acute exercise with different exercise modalities on cognitive flexibility in individuals with high and low aerobic fitness. Previous studies have shown that 10 min HIIE is similar to 20 min MICE in terms of energy expenditure and improving executive function, with the advantage of being more time efficient [24,35]. Using a between-subjects crossover design, 66 participants were divided into high- and low-fitness groups. All participant performed one 10 min HIIE session, one 20 min HIIE session, one 20 min MICE session, and one control session where they sat quietly and read for 24 min on separate days, in a counterbalanced sequence. A more-odd shifting task was performed before and after the acute exercise session to assess cognitive flexibility. Based on previous literature investigating cognitive flexibility [26,29], we hypothesized that there would be differences between the high and low fitness groups performing a cognitive flexibility task after performing acute exercises, compared to pre-exercise; and no differences in the control condition. Additionally, we also hypothesized that high-fitness participants would display better cognitive flexibility than low-fitness participants after acute exercise, and that the positive effect on cognitive flexibility would be greater after HIIE than MICE.

## 2. Materials and Methods

### 2.1. Participates

The sample size was determined based on a priori power (G*power 3.1.9.2) analysis with a large effect size (*η*^2^ = 0.19) [33], α level of 0.05 and a power (1 − *β*) of 0.80 at the group level, resulting in at least a sample size of 30 to detect similar significant effects. One hundred healthy college students were recruited from Qingdao University to participate in an aerobic fitness test. Participants completed the Physical Activity Readiness Questionnaire (PAR-Q) [36] to confirm their ability to safely engage in exercise. Prior to the formal experiment, all participants were required to perform a 20 m Progressive Aerobic Cardiovascular Endurance Run (PACER) test to determine the aerobic fitness level of all participants. Based on the estimated the maximum oxygen consumption (VO_2max_) of participants in the PACER test, the top third and bottom third of male and female participants were split into the high-fitness (*n* = 33) and low-fitness (*n* = 33) groups, respectively. The inclusion criteria were as follows: (1) right-hand dominant; (2) normal or corrected-to-normal vision and without color blindness; (3) no neurological, cardiovascular, or pulmonary disorders were reported; (4) no history of drug intake that affected cognitive function. These criteria were set to avoid the impact of moderators on the results of the experiment. Written informed consent was obtained from all participants. Participants were reminded not to drink alcoholic or caffeinated beverages and to have sufficient sleep within 24 h of study participation. The study protocol was approved by the Medical Ethics Committee of the Affiliated Hospital of Qingdao University and was conducted according to the ethical requirements of the Declaration of Helsinki. Final demographic characteristics and fitness data for the two groups are provided in Table 1.

### 2.2. Procedure

A between-subjects crossover design protocol was employed. Participants were asked to visit the laboratory five times. During the first visit, participants provided informed consent, demographic information, completed the PAR-Q, provided resting heart rate (RHR) and GXT measurements, and were instructed to practice the more-odd shifting task until they achieved an accuracy of at least 85%. Their aerobic fitness was assessed by the 20 m Progressive Aerobic Cardiovascular Endurance Run (PACER) test. On their second to fifth visits, participants completed a 10 min HIIE session, a 20 min HIIE session, a 20 min MICE session, or a control session in a counterbalanced order to avoid experimental sequence effects. In addition, participants completed the more-odd shifting task before (pre) and after (post) each session. All exercise sessions were completed on a treadmill (ICON 705CST). The perceived exertion scale (RPE) was used to assess subjective exertion [37], and was rated every 5 min during exercise. Participants were paid 150 RMB after completing the entire protocol. The study design is illustrated in Figure 1A.

### 2.3. Aerobic Fitness Assessment

Young adults’ aerobic fitness was assessed using the 20 m PACER test [38]. Subjects have to run back and forth on a 20 m course and touch the 20 m line with their foot, and at the same time, a sound signal is emitted from a prerecorded tape. The frequency of the sound signal increases by 0.5 kph every minute, indicating the next stage (level) and starting with a speed of 8.5 kph. The test ends when subjects fail to reach the line before the signal. The higher the number of laps a participant completed, the higher the aerobic fitness level they had. Maximal oxygen uptake (VO_2max_; mL·kg^–1^·min^–1^) was estimated from the number of the last stage reached as [−24.4 + 6.0 × velocity]. The PACER test is the most widely used field-based measure of aerobic fitness, demonstrating high reliability and validity in children and youth [39].

### 2.4. Exercise Protocols

Each participant underwent a GXT to determine maximum heart rate (HR_max_) using a treadmill [40]. The starting velocity and incline was set at 8.5 km/h and 3%, respectively. The incline was held constant while the speed of the treadmill increased by 0.5 km/h per minute until the participant reached volitional exhaustion as defined by at least two of the following three criteria: (a) a plateau in heart rate resulting in no increase with increased workload, (b) a peak HR ≥ age-predicted HRmax (208 − (0.7 × age)) [41], and (c) RPE ≥ 17. Resting heart rate (RHR) was obtained before the GXT using a Polar H10 heart rate strap (Polar, Kemple, Finland) while the participant was seated quietly. Heart rate reserve (HRR) was calculated by subtracting RHR from HR_max_ and was used to determine exercise intensity.

During the 10 min HIIE sessions, participants completed 5 bouts of repeated 1 min runs on a treadmill at an intensity focusing on 90% HRR (90% HRR + RHR), interspersed with 1 min of self-paced walking at 50% HRR (50% HRR + RHR) [42]. The 20 min HIIE intervention condition contained 10 bouts of repeated 1 min runs on a treadmill at an intensity focusing on 90% HRR, interspersed with 1 min of self-paced walking at 50% HRR. During the 20 min MICE session, participants finished 20 min of running on a treadmill at an intensity of 40% to 59% HRR (40–59% HRR + RHR). Each exercise started with a 2 min warm-up and finished with a 2 min cool-down. During the control session, participants read a book quietly for 24 min while sitting on a chair. All participants wore a Polar H10 heart rate belt throughout sessions. The exercise protocols are illustrated in Figure 1B.

### 2.5. Task-Switching

Cognitive flexibility was assessed by the more-odd shifting task [43]. The task was produced by a computer program utilizing E-Prime 2.0 (Psychology Software Tools Inc., Pittsburgh, PA, USA). Participants were seated in a quiet room at approximately 80 cm from a 15.6-inch monitor. The ck number (1–9, excluding 5) was displayed at the center of the computer screen. Each digit was displayed for 2000 ms and separated by 1000 ms inter-stimulus intervals. The task consisted of two conditions: non-switching (block A and block B) and switching (block C). Block A involved 8 non-switching trials during which participants were asked to identify whether the black number was greater or less than 5. Block B involved 8 non-switching trials in which the participants were asked to identify whether the green number was odd or even. Block C consisted of 16 switching trials in which participants were asked to determine the magnitude of the digits in black and the parity of the digits in green. The blocks were counterbalanced in an ABCCBA sequence to minimize order effects. Participants were instructed to press the “F” key with their left index finger as accurately and quickly as possible if the number in black was smaller than 5 or the number in green was even, and press the “J” key with their right index finger if the black number was larger than 5 or the green number was odd. The presentation of each stimulus was terminated when participants responded to it, or when a maximum stimulus presentation (2000 ms) was reached. The total number of trials is 64 and the task lasts approximately 5 min. The accuracy and response time (RT) of each response was recorded. Accuracy and the mean RT from response-correct trials were calculated for each condition. In addition, switch cost was calculated by RTs (switching condition)–RTs (non-switching condition) [44]. Smaller switch cost indicated better cognitive flexibility.

### 2.6. Statistical Analysis

For RT analysis, incorrect trials (7.73%) were first removed, and trials with an RT below 200 ms (0.01%) or above 2 standard deviations (SD) (4.70%) from the mean for each condition (switching and non-switching) were discarded. A preliminary analysis was conducted to examine the experiment session order, sequence effects were not found. Response accuracy and RTs were analyzed using a 2 (fitness: high and low) × 4 (session: 10 min HIIE, 20 min HIIE, 20 min MICE, and control) × 2 (time point: pre and post exercise) × 2 (task condition: switching and non-switching) repeated-measures analysis of variance (RM ANOVA). Switch cost was analyzed using a 2 (fitness: high and low) × 4 (session: 10 min HIIE, 20 min HIIE, 20 min MICE, and control) × 2 (time point: pre and post) RM ANOVA. Differences in demographic characteristics and fitness data between the high- and low-fitness groups were analyzed with independent sample *t*-tests. Mauchly’s test was used to examine spherical data, and the Greenhouse–Geisser correction was used to analyze non-spherical data. The Shapiro–Wilk normality test was applied to confirm the normal distribution of data. Non-normal data were logarithm transformed to correct for deviations from normality. *T*-tests with Bonferroni adjustments for multiple comparisons were applied for post hoc analyses. The statistical significance level for post hoc analyses was set at *p* < 0.00625 for analysis of response accuracy and RTs (eight comparisons were tested within each fitness group) and *p* < 0.0125 for analysis of switch cost (four comparisons were tested within each fitness group). Effect sizes are presented as partial squared (*η*^2^) values. Statistical analyses were performed using the Statistical Package for Social Sciences software (SPSS version 25.0, Chicago, IL, USA).

## 3. Results

Preliminary analyses did not reveal any interactions involving session order, (*F*_(3,195)_ = 1.95, *p* = 0.280, *η*^2^ = 0.02). Thus, further analyses were collapsed across session order.

### 3.1. Reaction Time

The four-way ANOVA on RT revealed a significant interaction effect of fitness × task condition × session × time point (*F*_(3,192)_ = 4.00, *p* = 0.009, *η*^2^ = 0.06). The post hoc test showed that in the high fitness group, reaction times were significantly lower after 20 min HIIE compared to before 20 min HIIE in the switching condition (*ps* < 0.001). In the low fitness group, reaction times after 10 min HIIE and 20 min MICE were significantly lower compared to pre-interventions (*ps* < 0.001) in the switching condition. There were no significant differences before and after the four interventions in the non-switching condition for either the high or low fitness group after Bonferroni correction (*ps* ≥ 0.009). No significant difference was found in the response time after reading intervention (*ps* ≥ 0.181) in either high- or low-fitness groups (Figure 2).

There was a significant main effect of time point (*F*_(1,64)_ = 68.54, *p* < 0.001, *η*^2^ = 0.52) on RT, with significantly shorter RT after the intervention (598.47 ± 118.30 ms) compared to before the intervention (620.99 ± 125.37 ms). There was a significant main effect of task condition (*F*_(1,64)_ =380.70, *p* < 0.001, *η*^2^ = 0.86), with longer RTs in the switching condition (674.84 ± 121.82 ms) compared with the non-switching condition (544.62 ± 81.44 ms). No significant main effect of fitness was observed (*F*_(1,64)_ =3.08, *p* = 0.084, *η*^2^ = 0.05). Mean accuracies and RTs are shown in Appendix A.

### 3.2. Switch Cost

There was a significant interaction effect of fitness × session × time point (*F*_(3,192)_ = 4.00, *p* = 0.009, *η*^2^ = 0.06) on switch cost. The post hoc test revealed that the switch cost in the high-fitness group was significantly lower after 20 min HIIE than before exercise (*p* = 0.001), with no significant changes after 10 min HIIE, 20 min MICE or control interventions (*p* ≥ 0.179). For the low-fitness group, the switch cost was significantly lower after the 10 min HIIE (*p* < 0.001) compared to pre-session. There were no significant changes in switch cost after 20 min HIIE, 20 min MICE or the control condition compared to before the intervention after Bonferroni correction (*ps* > 0.037).

There was a significant main effect of time point (*F*_(1,64)_ = 18.62, *p* < 0.001, *η*^2^ = 0.23) on switch cost with lower switch cost post-intervention (120.61 ± 74.55 ms) compared to pre-intervention (139.83 ± 78.08 ms). There was a significant main effect of fitness on switch cost (*F*_(1,64)_ = 4.32, *p* = 0.042, *η*^2^ = 0.06) lower switch cost in the high-fitness group (116.35 ± 69.20 ms) than that in the low-fitness group (143.82 ± 81.68 ms). No significant main effect of session (*F*_(3207)_ = 1.42, *p* = 0.237, *η*^2^ = 0.02) was observed (Figure 3). Mean switch costs are shown in Appendix A.

### 3.3. Accuracy

The four-way ANOVA on accuracy revealed no significant interaction effect of fitness × task condition × session × time point (*F*_(3,192)_ = 1.605, *p* = 0.190, *η*^2^ = 0.02). Neither the three-way interaction of fitness × session × time point (*F*_(3,192)_ = 0.161, *p* = 0.923, *η*^2^ = 0.003) nor the two-way interaction of session × time point (*F*_(3,192)_ = 0.907, *p* = 0.439, *η*^2^ = 0.01) was significant.

There was a significant main effect of task condition (*F*_(1,64)_ = 71.82, *p* < 0.001, *η*^2^ = 0.53), with greater mean accuracy during the non-switching condition (93.25 ± 5.16%) compared to the switching condition (90.60 ± 5.85%, *p* < 0.001). However, there were no significant main effects of session (*F*_(3,192)_ = 0.63, *p* = 0.595, *η*^2^ = 0.01), time point (*F*_(1,64)_ = 3.39, *p* = 0.070, *η*^2^ = 0.05), or fitness (*F*_(1,64)_ = 0.81, *p* = 0.372, *η*^2^ = 0.01) on accuracy.

### 3.4. Acute Exercise Performance

The interaction between fitness × session (*F*_(3,192)_ = 2.04, *p* = 0.109, *η*^2^ = 0.96) on HR was not significant. There was a significant main effect of session on mean HR (*F*_(3,192)_ = 2488.00, *p* < 0.001, *η*^2^ = 0.98). As expected, the mean HR during the 10 min HIIE (161.97 ± 7.40) and 20 min HIIE (163.97 ± 8.45) were both significantly greater than that during the 20 min MICE (137.75 ± 6.30) and control conditions (71.82 ± 8.61, all *p* < 0.001). Additionally, the mean HR during the 20 min MICE was significantly greater than that during the control condition (*p* < 0.001). There was no difference in mean HR between 10 min HIIE and 20 min HIIE (*p* = 0.583). There was a significant main effect of fitness on HR (*F*_(1,64)_ = 923.87, *p* = 0.001, *η*^2^ = 0.16) with a significantly greater HR in the low-fitness group (135.75 ± 39.25) compared to the high-fitness group (132.01 ± 37.05).

Analysis of the RPE scores showed no significant interaction between fitness × session (*F*_(1,128)_ = 1.85, *p* = 0.162, *η*^2^ = 0.03) or main effect of fitness (*F*_(1,64)_ = 3.10, *p* = 0.083, *η*^2^ = 0.05). There was a significant main effect of session (*F*_(2,128)_ = 238.55, *p* < 0.001, *η*^2^ = 0.82), with significantly higher RPE scores after 20 min HIIE (17.38 ± 1.57) compared to 10 min HIIE (15.58 ± 1.75) and 20 min MICE (12.27 ± 1.58) (all *p* < 0.001) on RPE score.

## 4. Discussion

The current study examined the cognitive flexibility of high- and low-fitness young adults using a cognitive shifting task before and after acute exercise (10 min HIIE, 20 min HIIE, and 20 min MICE) and a control condition (reading quietly in a chair for 24 min). Results revealed that aerobic fitness moderated the effect of acute HIIE and MICE on cognitive flexibility, with high-fitness participants deriving greater improvement in cognitive flexibility from 20 min HIIE, and low-fitness participants benefiting more from 10 min HIIE and 20 min MICE.

The present study demonstrated that cognitive flexibility of the high-fitness group, but not the low-fitness group, was improved after 20 min HIIE. This result is in agreement with previous studies that examined other aspects of executive function, including inhibitory control and working memory [45,46]. For example, Cooper et al. (2018) used the Stroop and Sternberg tasks to assess the effects of 60 min HIIE-based basketball training on inhibitory control and working memory, and found that high-fitness adolescents benefited more from immediately after exercise than low-fitness adolescents [46]. Similarly, Williams et al. (2020) found that only high-fitness participants demonstrated an improvement in working memory immediately after 60 min of an HIIE-based soccer session [45]. Although the study designs and exercise intervention differed from the current study, most importantly the same high-intensity intermittent exercise was used. The influence of the duration of exercise may depend upon an individual’s tolerance and familiarity with the physical activity. Twenty minutes of HIIE may be relatively easy to complete for a high-fitness individual who regularly engages in a long bout of activity, but may be quite difficult and hence too demanding for low-fitness individuals [47]. Indeed, the American College of Sports Medicine (ACSM) recommendation for effective aerobic training is fitness-level related, with greater HRR for fit individuals (80–85%) than for unfit individuals (50–60%) [40]. There is much evidence that exercise induces an increase in brain-derived neurotrophic factor (BDNF) which may be a possible mechanism for improved cognitive flexibility [48]. According to Dinoff et al. (2017), greater increases in BDNF after acute exercise were observed in those with higher aerobic fitness [49]. This may indicate that individuals with greater aerobic fitness are better ‘primed’ for acute physiological changes occurring after exercise and may receive greater acute benefits to cognition. A systematic review indicated that most high-intensity interval exercise sessions have positive effects on executive functions [50]. The current study confirms this and further suggests that only high-fitness participants may derive cognitive benefits from 20 min of HIIE.

The present study discovered that 20 min MICE improved cognitive flexibility in low-fitness but not high-fitness participants in switching condition. This finding supports several recent studies reporting improvements in cognitive performance in low-fitness participants after MICE [33,51,52]. For example, Cui et al. (2020) used the Stroop task to assess executive function in individuals with different aerobic fitness before and after MICE, and found a significant decrease in response time after acute aerobic exercise [34]. The results of the post hoc analysis further indicated that this positive effect mainly manifested in the low-fitness group rather than the high-fitness group. Similarly, Li et al. (2019) assessed college students’ performance on the N-back task before and after 20 min MICE and observed significantly better task performance in the 0-back task after exercise than before exercise in the low-fitness group [33]. However, there was no significant difference in task performance before and after exercise in high-fitness group. Conversely, several other experimental studies have suggested that high-fitness individuals derive more benefit from acute aerobic exercise [30,31]. Differences in criteria used to classify aerobic fitness level may contribute to the discrepancy in these results. Studies using strict differentiation criteria to classify aerobic fitness have shown that the low-fitness group benefits from MICE. For example, Cui et al. [34] classified participants in the top or bottom 27% (group mean) of the rankings as high or low fitness based on the assessment of aerobic fitness levels, and found that found that the low-fitness group obtained greater acute exercise benefits than the high-fitness group. Conversely, cognitive performance after MICE was found to be superior for high-fitness participants when individuals in the top or bottom 50% of fitness test scores were divided into high- or low-fitness groups [31,53].

The current study also found that 10 min of HIIE improved cognitive flexibility more in low-fitness participants relative to high-fitness participants. Kao et al. found that 9 min of HIIT had a positive effect on inhibitory control in young adults with high aerobic fitness. However, only higher fitness young adults were involved in their study and it was not possible to determine whether HIIE of similar duration was more beneficial for cognitive performance in high or low fitness young adults. A meta-analysis suggested that the duration of the exercise session influenced the effect of acute exercise on cognitive performance, with shorter exercise bouts having a negative impact on cognitive performance and longer exercise bouts having a positive effects [11]. Thus, when assessed post exercise, it appears that at least 20 min of exercise is necessary to see effects. It is a new finding in the current study that only 10 min of HIIE can improve cognitive flexibility, but only in low-fitness individuals. In contrast to the current study, some previous studies that used 10 min MICE as an intervention to examine changes in cognitive performance before and after exercise found no benefit on post-exercise cognitive performance [31]. The differential results of the study were not only related to the population of the exercise intervention, but they may also be closely related to exercise intervention used.

Acute exercise-induced cortical activation (e.g., prefrontal cortex, PFC) is a possible explanation for the disproportionate benefit of HIIE and MICE on participants’ cognitive effects. Indeed, an individual’s aerobic fitness level can modulate the perception of activities induced by exercise stimuli or physiological demands [47,54]. Brain processes that enhance executive function appear to respond differently to exercise intensity, which may be related to the sensitivity to physiological stress in people with different levels of aerobic fitness. PFC activation follows an inverted U-shaped curve such that a moderate acute stressor, via glucocorticoid receptor activation, can positively modulate PFC-mediated cognitive processes by enhancing glutamate receptor trafficking and excitatory synaptic transmission in this region [55], whereas a severe stressor can impede this process [56]. 20 min HIIE consumes more energy and has a larger measure of physiological stimulation relative to 20 min MICE. Moreau and colleagues suggest that there may be a negative correlation between duration and intensity, with shorter duration compensating for the larger effect associated with higher intensity [22], which could explain the 10 min HIIE and 20 min MICE comparable positive benefits. Thus, it is reasonable to speculate that high-fitness young adults may have higher tolerance and familiarity with exercise stimuli than low-fitness individuals, and that 20 min of MICE or 10 min of HIIE is a moderate stressor for low-fitness young adults and a 20 min HIIE for high-fitness individuals, respectively, to promote optimal activation of PFC.

The current study has some limitations that should be considered when interpreting the findings. Firstly, this study only verbally prompted participants to pay attention to sleep, dietary intake, and physical activity, and future studies should use relevant devices for monitoring and strict control of variables. Secondly, we did not take heart rate measurements after exercise; thus, it is unclear whether participants’ heart rate had returned to 10% of resting values. Future research is needed to explore how the timing of post-exercise cognitive assessment modifies the effect of acute aerobic exercise on cognitive performance. Finally, this study did not match participants in higher and lower aerobic fitness groups in terms of intelligence quotient and socioeconomic status or control for these variables in the analyses. Given that intelligence and socioeconomic status are both predictors of cognitive control, more research is needed to see if differences in cognitive flexibility improvements between acute HIIE and MICE are influenced by intelligence or socioeconomic status.

## 5. Conclusions

The present study demonstrates that different modalities of acute exercise (10 min HIIE, 20 min HIIE, and 20 min MICE) influence cognitive flexibility in high- and low-aerobic fitness young adults differently. Overall, the results provide new evidence that aerobic fitness and exercise modalities together modulate the effects of acute exercise on cognitive flexibility in young adults. Specifically, young adults with a high fitness level showed greater enhancement in cognitive flexibility after 20 min HIIE, while individuals with a low fitness level benefited more from 10 min HIIE and 20 min MICE. College students who exercise are advised to take their fitness level into account when choosing the type of exercise that is most time-efficient and maximizes cognitive ability. These results should be publicly distributed to achieve greater efficiency in the workplace or at school.

## Figures and Tables

**Figure 1 ijerph-19-09106-f001:**
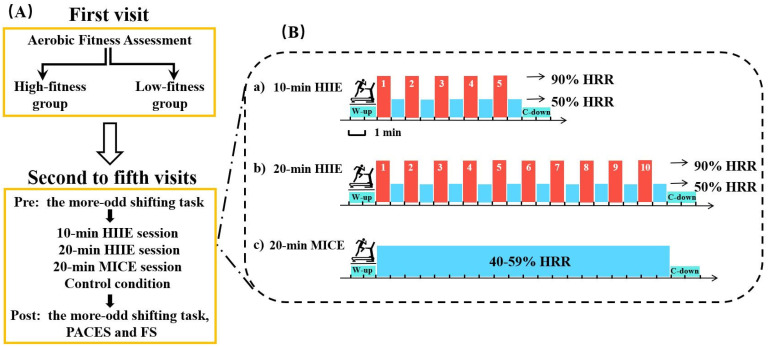
(**A**) the study design; (**B**) the experimental protocol. HIIE: high-intensity interval exercise; MICE: moderate-intensity continuous exercise; W-up: warm up; C-down: cool down.

**Figure 2 ijerph-19-09106-f002:**
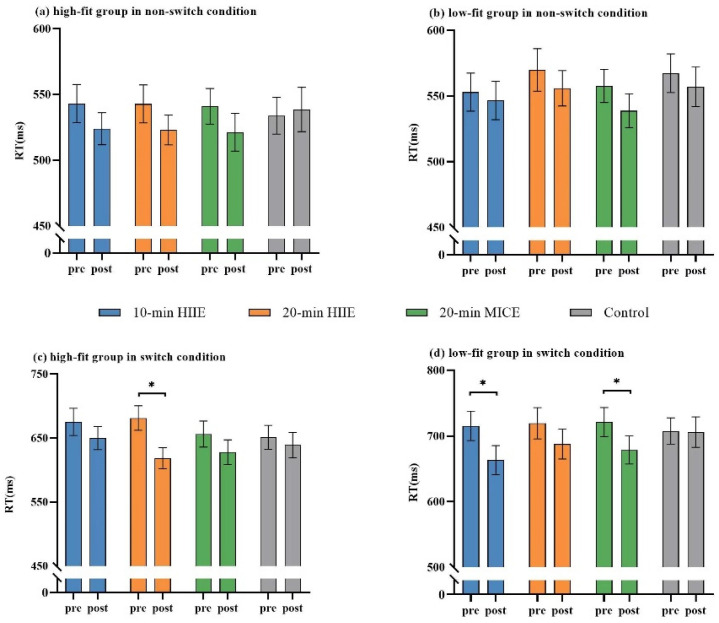
RTs under the non-switching condition in high-fitness group (**a**) and low-fitness group (**b**), and RTs under the switching condition in the high-fitness group (**c**) and the low-fitness group (**d**). Error bars represent the standard error for the mean (* *p* < 0.00625 for Bonferroni correction).

**Figure 3 ijerph-19-09106-f003:**
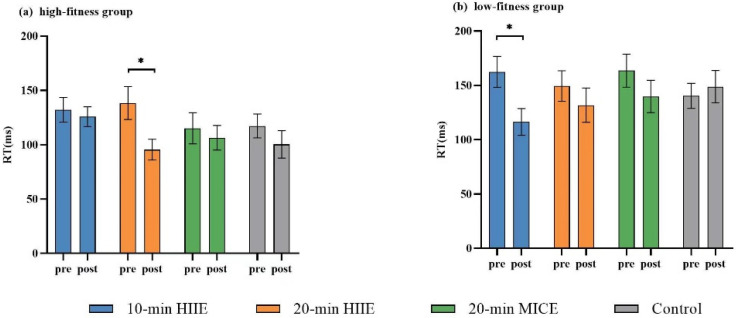
Switch cost for the high-fitness group (**a**) and low-fitness group (**b**) on the more-odd shifting task (* *p* < 0.0125 for Bonferroni correction).

**Table 1 ijerph-19-09106-t001:** Descriptive data of the high- and low-fitness groups (*Mean* ± *Standard Deviation*).

Variable	High-Fitness	Low-Fitness	*p*
Gender (male/female)	33 (16/17)	33 (16/17)	-
Age (years)	19.65 ± 1.05	19.29 ± 0.85	0.132
Height (cm)	174.24 ± 6.92	171.97 ± 8.58	0.240
Weight (kg)	63.91 ± 7.99	67.79 ± 21.13	0.327
BMI (kg/m^2^)	20.88 ± 1.60	23.23 ± 4.94	0.011 *
Body fat (%)	19.02 ± 5.12	23.93 ± 5.65	<0.001 ***
HR_max_ (bpm)	192.91 ± 7.72	192.18 ± 7.88	0.706
HRR (bpm)	131.03 ± 10.35	123.30 ± 10.82	0.004 **
VO_2max_ (mL·kg^–1^·min^–1^)	56.35 ± 7.01	35.05 ± 4.27	0.001 ***
Mean 10 min HIIE HR (bpm)	159.14 ± 7.82	164.79 ± 5.80	0.001 ***
Mean 20 min HIIE HR (bpm)	161.55 ± 7.99	166.39 ± 8.31	0.019 *
Mean 20 min MICE HR (bpm)	135.58 ± 6.07	139.92 ± 5.84	0.004 **
Mean HR during rest (bpm)	71.76 ± 6.55	71.89 ± 10.37	0.951
10 min HIIE RPE	15.42 ± 1.60	15.73 ± 1.89	0.485
20 min HIIE RPE	17.15 ± 1.73	17.61 ± 1.37	0.242
20 min MICE RPE	11.73 ± 2.04	12.82 ± 1.79	0.024 ^*^

Note: BMI = body mass index; RPE = ratings of perceived exertion; HR = heart rate; HRR = heart rate reserve; * *p* < 0.05, ** *p* < 0.01, *** *p* < 0.001.

## Data Availability

The data presented in this study are available on request from the corresponding author.

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
