# Peer review of "Effects of Acute Exercise on Cognitive Flexibility in Young Adults with Different Levels of Aerobic Fitness"

_ijerph, 2022, doi:10.3390/ijerph19159106_

Round 1
Reviewer 1 Report
Hello.
I feel your work was very well done and interesting to read. Your paper was very well written and easy to follow. One minor suggestion
- Formatting of line 151
Good luck with your work.
Author Response
Thanks very much for taking your time to review this manuscript. We really appreciate all your generous comments and suggestions! Please find my responses in the resubmitted manuscript.
Reviewer 2 Report
Review
Title: Effects of Acute Exercise on Cognitive Flexibility in Young 2 Adults with Different Levels of Aerobic Fitness
Authors: Beibei Shi, Hong Mou, Shudong Tian, Fanying Meng, and Fanghui Qiu
Journal: International Journal of Environmental Research and Public Health
Comments to Authors
Review date: 05/12/2022
Overview
The manuscript by Shi et al. reports a randomized cross-over study on the moderating effects of aerobic fitness on cognitive flexibility assessed with a computerized more/odd switching task. The study employed 4 conditions, including a 10 min high-intensity interval exercise (HIEE), 20 min HIEE, 20 min of moderate-intensity continuous exercise (MICE), and quietly reading for 24 min counterbalanced conditions. The study was conducted on a sample of college students, and the results suggested a beneficial effect of a single bout of MICE and 10-min HIIE on a global switch cost. In contrast, high-fit individuals only benefited from 20-min HIEE.
While this is an interesting study that would enhance our understanding of aerobic fitness in individual responses to acute bouts of physical activity of varying intensity, several methodological flaws dampen this reviewer's enthusiasm. The major and potentially fatal flaw of the study is their assessment of aerobic fitness. It is unclear to this reviewer which validated protocol was used to assess aerobic fitness and why VO2max was not estimated based on a validated equation. The currently reported method does not measure aerobic fitness and precludes comparison of the sample to published international norms to define aerobic fitness groupings. Unless the authors can address this issue, the group split based on what authors call aerobic fitness (without providing details other than aerobic fitness groups were split based on the tertiles "according to the results of the graded exercise test") are invalid. Without the moderating effect of aerobic fitness, this manuscript is largely a replication of the previous study by Tien et al. 2021 except for adding a 10-min HIIE condition which was previously done in relation to inhibitory control (see Kao et al. 2017). Thus, it would be challenging to argue novelty based on the addition of this condition alone.
Major Comments
Abstract
Methods to define aerobic fitness groups should be reported. The timing of cognitive assessment should be included in the abstract.
Introduction
Terminology in the introduction is vague in relation to "exercise." The authors study the effects of an acute bout of aerobic physical activity and HIIT on cognitive flexibility. Exercise is defined as planned, structured, and repetitive physical activity to improve or maintain physical fitness (Caspersen et al., 1985). Pontifex et al. 2019 have drawn attention to correct terminology in what is commonly known as "acute exercise" literature. If authors refer to "exercise," at the very least, they ought to specify which exercise modality they are referring to (e.g., lines 28, 29, 41). Authors also need to define which exercise modalities they refer to by "diverse exercise modalities" (line 31). These details matter because differences in exercise modalities are associated with differences in hypothesized mechanisms underpinning the effects of acute bouts of physical activity on cognitive functions (e.g., an increase in neurotrophic factors compared to neural stimulation by cognitive complexity of movement tasks).
In lines 36-37, the last sentence of para 1, the authors do not explain how exercise modality and aerobic fitness levels modulate the effects of acute (aerobic?) exercise on cognitive functions. The exercise modalities included in the study also vary in exercise intensity. Based on the current design, the effect of exercise modality cannot be dissociated from that of exercise intensity. Specifically, there is no continuous vigorous exercise condition, which could be compared to an HIIE condition. This reviewer would suggest that authors explicitly discuss the differences in the expected effects of exercise modality on cognitive performance. For example, in the meta-analysis of the acute effects of aerobic physical activity and high-intensity interval exercise (HIIE) on inhibitory control, Oberste et al. 2019 found no significant differences in the effect sizes between HIIE and vigorous intensity physical activity. Thus, the intensity rather than the modality (when HIIE is compared to a bout of continuous aerobic physical activity) is an equally likely explanatory factor in the observed effects of MICE and HIIE on task switching.
In lines 67-71, the authors need to explain how and why aerobic fitness classification could lead to discrepant findings. This is partly explained in the discussion, but this argument should be elaborated upon in the introduction to provide a rationale for the moderating effect of aerobic fitness on the acute effects of HIIE and MICE on task switching. The predictions based on the aerobic fitness hypothesis are not clearly supported by evidence. In their meta-analysis, Chang et al. 2012 found that the overall effect of an acute bout of physical activity on cognitive performance was positive in individuals with high and medium aerobic fitness when cognitive functions were tested at a delay after a physical activity bout. When tested immediately after a physical activity bout, physical activity also did benefit those with low aerobic fitness. To ignore these discrepancies in the evidence base and take the cardiovascular fitness hypothesis at its face value can lead to incorrect hypotheses. Thus, authors should review the evidence to support their rationale for the greater effects of an acute bout of MICE and HIIE on task switching in lower fit individuals.
Likewise, the evidence on the effects of intensity of acute bouts of physical activity on cognitive functions is mixed. Chang et al. 2012 showed comparable effect sizes for physical activity bouts of light, moderate, and hard intensity, with only very hard intensity showing a larger effect size. Moreau et al. 2019 meta-analyzed studies testing the acute effects of HIIE on cognitive functions and comparing HIIE to lower-intensity (moderate and light) conditions. They found no statistically significant effect sizes for the difference between HIIE and lower-intensity conditions. Furthermore, when the intensity of high-intensity bouts in the HIIE condition was assessed as a moderator, only the lowest intensity of high-intensity intervals (77-88.5% of HR max) yielded a statistically significant effect. Both these meta-analytical findings do not support the drive theory.
Studies by Tsai et al. and Netz et al. also differ in their studied populations; Tsai et al. included young adult males only, while Netz et al. studied older and younger women. Thus, the differences in cognitive performance between aerobic fitness groups in the latter study were confounded by both age and physical activity levels. Although the authors reported that aerobic fitness specifically predicted pre-to-post change in cognitive flexibility, this result was based on correlational evidence. Thus, the discrepancies between these two studies are not a good representation of discrepant findings on the moderating effects of aerobic fitness on the effects of acute aerobic exercise on cognitive control.
Methods
Participants
Were the participants screened for the use of antidepressants and other medication that could affect cognitive function? Screening based on drug or alcohol use does not imply screening based on medication affecting neurological and cognitive functions.
The timing of the testing sessions has not been reported, although the authors state that each session was completed at approximately the same time of the day. The lack of specific information on the average time of pre and post cognitive assessments for each condition is a limitation due to diurnal changes in cognitive performance.
Likewise, the authors state that participants were "reminded not to drink alcoholic or caffeinated beverages and to have a sufficient sleep within 24 hours of study participation". However, these factors were not measured. Thus, it is impossible to ascertain if differences in sleep, caffeine, and alcohol consumption did not contribute to the observed moderating effects of aerobic fitness. The lack of physical activity monitoring prior to experimental visits is also a limitation due to its effects on physiological responses to aerobic exercise.
Experimental design
The methods lack a description of whether a design was balanced and how the statistical analyses accounted for sequence and period effects. This is important because unbalanced designs reduce the efficiency of estimation of the direct effects (and carry over effects) and may affect study conclusions, for example, concerning accuracy. While carry-over effects are likely not a concern in this study due to sufficient wash-out period, practice effects could lead to sequence effects. Please explicitly state whether a balanced design was used and if carry-over effects were tested but were not present.
What were the average times post-exercise of cognitive assessments? Please include a measure of dispersion and minimum and maximum times. This information is important because the timing of post-exercise cognitive assessment has been shown to modify the effect of acute aerobic exercise on cognitive performance (see e.g. Chang et al., 2012). Likewise, the authors do not specify whether the cognitive performance was tested after the heart rate had returned to 10% of its resting value. Thus, the differences between cognitive responses to exercise between higher and lower fit individuals could reflect the differences in physiological arousal and subjective experiences of physiological arousal and physical effort. This interpretation is supported by differences between aerobic fitness groups in their ratings on the feeling scale and physical activity enjoyment scale. The potential effect of the timing of cognitive assessment post-exercise on the moderating effects of aerobic fitness on cognitive responses to MICE and HIIE protocols should be discussed and considered a limitation if the HR was not allowed to return to 10% of its resting value. Please report the HR values reached before the cognitive performance was tested.
Cognitive task.
The differences in cognitive improvements following MICE and HIIE conditions between aerobic fitness groups could also be related to the differences in performance variability. Lower aerobic fitness has been linked to greater variability in cognitive performance in pediatric populations (see for examples: Moore et al. 2013, Raine et al. 2018, Wu et al. 2011). Our data also suggest that engagement in higher intensity daily physical activity (a correlate of aerobic fitness) is related to smaller RT variability in older adolescents nearing the sample's age in this manuscript. Kamijo et al. 2019 found a decrease in variability in the RT on the nontarget trials of the 2-back 30 min after exercise cessation, which coincided with decreased RT relative to both baseline and assessment immediately post-exercise (a continuous 20 min of moderate intensity physical activity). Thus, it is plausible that the lack of pre-to-post differences in non-switch RT for the lower fit group was the function of increased variability in RT immediately post exercise (MICE And HIIT). This reviewer would recommend that the authors present findings on the effects of the intervention on the standard deviation and the coefficient of variation of the RT to help readers better understand the observed effects.
The analysis of the task switching task is incomplete. The authors assessed the global switch cost but did not examine the local switch cost (switch – non-switch trials in the mixed block).
VO2max measure
This reviewer has not come across a similar estimate of aerobic fitness based on criteria specific to a cardiopulmonary exercise test (CPET) protocol where CPET was not performed. Thus, the criteria used to determine VO2max from a CPET as per the ACSM Guidelines for Exercise Testing and Prescription, 11th edition, should not be used because authors have no means to verify their criteria against changes in oxygen uptake or Respiratory Exchange Ratio.
The authors describe a maximal exercise test where an individual's VO2max should be estimated from an equation. Unfortunately, the authors do not provide information on what specific protocol was used. To this reviewer's knowledge, the described protocol does not match any of the commonly described protocols for which VO2max prediction equations have been published (e.g., see Biles (ed.) 2018. The ACSM's testing and prescription. Philadelphia: Wolters Kluwer) and which are commonly used in physical activity and cognition literature (e.g., a modified Balke or a modified Bruce protocol). The authors seem to be avoiding the issue of the lack of VO2max estimate by stating that groups were split based on tertiles "according to the results of the graded exercise test" but do not disclose what results specifically they were referring to. Since VO2max was not measured directly or estimated, the authors cannot claim that their groupings represent individuals with high and low fitness. Basing high and low aerobic fitness groups on tertiles of the sample also precludes the generalizability of study findings to a broader population. Unless the authors can provide reference to a validation study of their GXT protocol and associated predictive equation for a VO2max and split aerobic fitness groups based on published international norms, this reviewer thinks that the study is unable to determine the differences in cognitive response to MICE and HIIT depending on the levels of aerobic fitness.
Unfortunately, without such analyses, the study is largely a replication of a similar study by this group published in IJEPRH (Tian et al., 2021; 18:9631) except for the addition of a 10-min HIIT condition. As such, the novelty of this contribution without the analyses based on aerobic fitness groups would be questionable.
Reference #41 (Ferguson et al. 2014) cited to support GXT termination criteria refers to a review of the 9th edition of the ACSM guidelines (https://www.ncbi.nlm.nih.gov/pmc/articles/PMC4139760/) not the guidelines themselves. The citation also refers to outdated guidelines (9th edition) since the 11th edition of the ACSM guidelines has already been published in 2022.
Covariates
This reviewer is also concerned about the lack of matching participants in higher and lower aerobic fitness groups on intelligence quotient and socioeconomic status or controlling for these factors in the analyses. Intelligence and socioeconomic status are correlates of cognitive control (including task switching) and important covariates to consider. Both factors are commonly included in the analyses of physical activity and cognition literature. Even though the included sample of college students may seem homogenous, considerable inter-individual variability in these factors is also prevalent among college students.
In addition, adiposity has been negatively related to cognitive performance (see work by Raine et al., Khan et al., and a meta-analysis by Yang et al., 2018). The groups differed in BMI and % total body fat mass. Thus, adiposity is a potential confounder of the reported findings, which were not accounted for.
Statistical analyses
Why were RTs below 300 ms discarded? Anticipatory latency is commonly defined as RTs below 100-200 ms (e.g., Jensen, 2006), and this definition has been widely applied in physical activity and cognition literature. Did these RTs all reach 2SD below the mean for each participant? Since RT distribution is negatively skewed, a cut-off based on the interquartile range rather than SD would be more appropriate.
Please provide power analyses to justify the sample size of the study. Multiple conditions in the reported study, the within-between group design, and the lack of controls for confounding effects of IQ and socioeconomic status leave the reviewer to wonder whether the study has the power to detect all hypothesized effects.
Please clarify if Bonferroni correction was applied and report the correct p-threshold for statistical significance. If the analyses were adjusted for multiple comparisons using the Bonferroni procedure, the p-value to denote the statistical significance of each comparison would need to be less than 0.05. If four comparisons were tested (pre- to post change within each fitness group per cognitive outcome), the p-value should be < 0.0125.
Results
Were the differences between aerobic fitness groups in the effect sizes comparing the change in pre-to-post assessment statistically significant?
Section 3.1: The authors report a 4-way interaction with task condition as the last factor in the interaction term. However, they split their subsequent analyses by task condition and conducted subsequent analyses within each condition. If this was the goal, then the condition should be entered first as a term in the 4-way interaction. The next step in decomposing the 4-way interaction would be to test a 3-way interaction between aerobic fitness, session, and within each task condition, followed by a two-way interaction of session by time within each aerobic fitness group if a 3-way interaction is statistically significant. However, the results of such interactions are not reported. Instead, the authors jump directly into fitness group comparisons of pre-to-post task performance. Then they go on to report two-way interactions across the entire sample regardless of aerobic fitness. This is confusing, as the authors already depicted the 4-way interaction splitting their analyses by task condition and jumping straight into the within aerobic fitness group comparisons. In this reviewer's view, it does not make sense to report 2-way interactions across the entire sample and task conditions if a 4-way interaction and 3-way interaction (?) were statistically significant because the interaction effect tells us that the effect of session and time will be different for each task and (assuming a 3-way fitness by a session by time interaction was statistically significant) aerobic fitness group.
The authors need to report the results of 3-way and 2-way interactions to support splitting their analyses by aerobic fitness group and session. At present this is not clear that these interaction effects were tested.
Section 3.2: Similar to my points listed under Section 3.1.: Given the statistically significant 3-way interaction, were session by time point effects interactions tested within each fitness group? The authors report no statistically significant session-by-time point interaction for the RT switch cost in the entire sample. However, they do not report the results of this 2-way interaction within fitness groups, which is surprising given the statistically significant 3-way interaction. Because we already know that fitness is a moderator, two-way interactions should be conducted within each fitness group. The lack of statistical significance for the session by timepoint only suggests no session effects on pre to post-change in cognitive performance across the entire sample. As this reviewer understands the analyses performed, this was not the point of conducting the moderating effect models with aerobic fitness as a moderator.
In general, results of the two-way interaction within each fitness group should be reported first, followed by the main effects. It does not make sense to report the significant main effect of aerobic fitness once we already know that it moderates the effect of sessions and time points.
This reviewer also recommends that authors compare the effect sizes of within-subject effects between aerobic fitness groups to statistically determine the magnitude of between-group effects.
Section 3.3: Please keep the order of reported analyses and terminology consistent across statistical models. The 3-way interaction should be reported first. If aerobic fitness was first entered into the model, followed by session and time point, this order should be reported in the interaction. The order of factors entered into the interaction term does not match that of the RT reported in section 3.1. Suppose a 4-way interaction of condition x aerobic fitness x session x time was not statistically significant. In that case, the 3-way interaction of aerobic fitness x session x time should be tested across task conditions, followed by a 2-way session by time interactions and main effects if these interactions were not statistically significant.
3.4. The effects of the interaction (aerobic fitness by session) should be reported first, followed by the main effects.
3.5. This section is unexpected. The authors do not report specific hypotheses relating the feeling scale and physical activity enjoyment to the differences in the acute effects of MICE and HIIT on task switching. Why are these results important? Explain how these results improve our understanding of the observed effects. Were these analyses included post hoc, or did the authors have an a priori hypothesis? At present, these results seem to be included ad hoc.
The results section seems to be assembled ad hoc. The narrative to guide the reader through the results justifying each step is missing. Please explain the reason for each step and what each step and what the analyses show us relative to these steps.
Discussion
Based on the design limitations, including the measure of aerobic fitness and group definition, the discussion will most likely change based on the re-analyses.
However, I provided general comments relating to the current discussion.
Firstly, the effect of exercise modality could not be parsed out from physical activity intensity. Thus, the summary of findings should clearly identify that the differences between conditions were based on physical activity intensity (moderate versus vigorous) and modality (MICE and HIIT) combined.
In the studies by Cooper et al. and Williams et al., when were the cognitive functions assessed? This is important to establish comparability of the findings with previous studies.
As noted above, Netz et al. studied older adult women. Because of age differences and sex differences in the samples, their study is not directly comparable to that reported in this manuscript. These differences should be recognized and a discussion of comparisons between the two studies qualified.
Lines 322-330: the discussion is vague on the details of aerobic fitness assessment group ascertainment in the discussed studies. Was the 27% bottom percent defined relative to a group mean or population norms? Notably, the authors cannot compare their results to previous studies because, based on their report, they did not estimate participants' aerobic fitness levels. See my comment under the methods section, VO2max assessment, for details.
The authors' interpretation of the similarities in arousal following MICE and 10-min HIIT protocols is not supported by their data. Specifically, the mean HR was higher in the HIIT condition compared to MICE conditions, suggesting higher arousal in the former. In addition, aerobic fitness did not moderate the HR responses to different conditions. Suppose the authors' argument of higher arousal explaining the comparable effects of MICE and a 10-min HIIT on switch cost were to hold. In that case, there should be no difference in HR between these two conditions in what the authors call a "low-fit" group.
The authors did not measure catecholamine response. Thus their interpretation of the results is speculative. They use language affirming this speculation: "The current study results suggest that 20-min MICE and 10 min HIIE were insufficient to trigger a beneficial response of the catecholamine level in high-fit individuals …". This and the following statements (lines 353-360) are incorrect. The study does not suggest that because no measures of catecholamines were taken. The authors may hypothesize a possible mechanism but cannot draw a conclusion without the supporting data.
The results relative to non-switch and switch trials are not discussed, and they should be considered in relation to which results in likely drive the switch cost.
Limitations
Limitations are severely understated. I have already pointed to the major limitations of the current study, including the lack of a valid estimate of aerobic fitness, statistical analyses (e.g., not accounting for the sequence and period effects), and the lack of adjustment for important covariates or better yet matching two fitness groups on these covariates. In addition, physical activity, sleep, and diet were not measured before each condition, and the timing of each condition and cognitive assessments is unclear. All these factors limit any firm conclusions that can be drawn from this study.
Minor Comments
Introduction
Methods
Line 90: Change "Participates" to "Participants"
A task could be described more clearly, including the number of blocks and the total number of trials used in the study. This is implied by the counterbalance order of task blocks but not explicitly stated.
Statistical analyses
The authors report that the Shapiro Wilk test was applied to confirm the normality of the data. Which data specifically? Were all outcomes normally distributed? Accuracy seems to be negatively skewed based on high % correct responses. Please confirm that all outcome variables were normally distributed as reported.
Results
A supplementary table including cognitive task performance for each session would be helpful to complement the figures and facilitate comparability with other studies.
Move section 3.4 to the beginning of the results section. This section informs the reader about the fidelity of the interventions.
References
References require formatting. For example, some article titles have all first letters in capitals, while others do not.
This reviewer recommends that the manuscript is proofread for multiple grammatical errors.
Author Response

(The authors gave the same response as above.)

Reviewer 3 Report
Overview
The authors aimed to investigate the effects of acute exercise with different modalities on cognitive flexibility in young adults with differing levels of aerobic fitness. The results suggested that aerobic fitness and exercise modalities may influence the effect of acute exercise on cognitive flexibility.
The paper is well written, and the topic is very interesting.
Here are my comments on the paper.
Specific comments
Abstract
It is of appropriate length, clear and understandable.
- Line 21, Keywords: Replace the following keywords "acute exercise; cognitive flexibility; aerobic fitness" with keywords other than those in the title. It is a suggestion for optimizing the search for the manuscript through search engines.
Introduction
The literature is summarized correctly; the authors clearly define the gap in the literature to be filled. The purpose and hypothesis are clear.
Materials and Methods
- Line 90: Replace with “2.1 Participants”
The methodology was clearly and correctly described.
The measurements of the dependent variables were carried out with Psychology software.
The statistical analyzes used are appropriate.
Results
Well written and correct.
Discussion
They are written clearly and to point.
The limitations are appropriate. It will be interesting to find out if the benefits obtained from acute exercise will be maintained over time.
The take-home message is clear.
Conclusions
The take-home message and future implications are clear.
Author Response

(The authors gave the same response as above.)

Round 2
Reviewer 2 Report
Please find my responses to the authors' responses attached.

Author Response
Dear Reviewer,
We appreciate you very much for your constructive comments and suggestions on our manuscript. We have carefully considered your comments and done our best to revise the manuscript accordingly. In this revised version, changes to our manuscript were all highlighted by blue-colored text.
We have done our best to revise our manuscript according to the comments. Due to time constraints, we have not yet requested professional touch-ups of the language in the manuscript. Please find the revised version attached, which we would like to submit for your kind consideration.
We have made detailed revisions in response to reviewer 2's comments. Regarding the delineation of aerobic fitness, we recaptured participants, tested them with a 20 m shuttle run, and assessed maximal oxygen uptake using a validated formula.
Once again, we appreciate the reviewers' comments on our manuscript, which allow us to continuously optimize the quality of the article. We will follow the reviewers' suggestions and study hard. And we hope that the corrections will meet approval. If you have any questions, please don’t hesitate to contact me at the address below.
Yours sincerely,
Prof. Fanghui Qiu
School of Physical Education, Qingdao University, 308 Ningxia Road, Qingdao, Shandong, China E-mail: qiufanghui@qdu.edu.cn
Postal Address: 308 Ning Xia Road, Qingdao 266071, China.
